# Diff-Fair: Mitigating Intersectional Bias Through Diffusion-Driven Fair Representation

## Abstract

Algorithmic fairness remains a critical challenge in Artificial-Intelligence, particularly for high-stakes domains where biased predictions can have significant societal consequences. While recent advances in fair representation learning have shown promise, existing approaches often struggle with the inherent trade-off between fairness and utility, especially when addressing intersectional fairness. In this paper, we introduce Diff-Fair, a novel diffusion-based framework that leverages the progressive denoising process to learn fair representation with the help of proposed fairness constraint for intersectional bias. Our approach simultaneously addresses multiple fairness dimensions through several complementary mechanisms, (1) a diffusion model for representation extraction, (2) a mutual information estimator to minimize sensitive attribute leakage in learned representation, (3) an intersectional fairness regularizer that explicitly accounts for overlapping demographic attributes, and, (4) a false positive rate regularizer that mitigates disparate impacts across groups. Through extensive experimentation on several real-world datasets from different domain, we demonstrate that Diff-Fair consistently outperforms state-of-the-art works, reducing demographic disparities and false-positive rate difference while maintaining competitive accuracy for both binary and multi-class classification.

## 1 Introduction

In recent years, Artificial Intelligence based systems increasingly influence critical decisions in domains such as healthcare, hiring, lending, and criminal justice, raising significant concerns regarding fairness and bias in these automated decision making systems Liu et al. (2022); Caton & Haas (2024). These concerns arise from the usage of historical biased dataset for training these AI-based models, which consequently exhibit bias in decision making process towards marginalized groups. A notable example occurred in 2018, when researchers found that a prominent healthcare algorithm used across U.S. hospitals showed systematic bias favoring white patients over dark-skin patients when allocating resources Obermeyer & Mullainathan (2019). This bias emerged because the employed algorithm used healthcare costs as a proxy for medical needs. Due to historical inequalities in healthcare access and spending, dark-skin patients with same severity of illness received lower risk scores than white patients. As a result, reliance on the model led to fewer resources being allocated to them Obermeyer & Mullainathan (2019). Findings like these highlight the necessity of building AI system that make fair decision even when learning from unfair historical data.

Recently, significant progress has been made in identifying and mitigating biases in datasets with respect to single sensitive attributes, e.g. race or gender, through technique like demographic parity constraints, adversarial bias mitigation, and post-processing methods Li et al. (2022); Rajabi & Garibay (2022); Liu et al. (2022). *Intersectional bias* arises when the conjunction of multiple sensitive attributes correlates with the outcome in a way that single-attribute bias-mitigation approaches fail to address. For example, a study of three commercial facial recognition systems revealed that they misidentified dark skinned women at a rate over 34 percent, while achieving near-perfect accuracy (less than 1 percent error) for light-skinned men Hardesty (2018). Notably, these systems *did* perform reasonably well when evaluated separately on gender or skin tone. This signifies the necessity

of models that is capable of maintaining high accuracy while simultaneously ensuring equitable outcomes across intersectional demographic subgroups.

In this paper, we introduce **Diff-Fair**, a comprehensive framework that **addresses intersectional bias through a multi-faceted approach, simultaneously targeting representation-level, group-level, and outcome-level fairness via several complementary mechanisms.** First, we employ a diffusion model as the representation learning backbone to create latent representation. Then, we design a dedicated classifier module that operates on these representations while incorporating fairness constraints directly into its training objectives. We also use a mutual information neural estimator to minimize the information leakage from the latent representation, thereby mitigating bias at the group level. Furthermore, we introduce an intersectional fairness regularizer that minimizes representational differences across intersectional groups. This approach directly addresses representational bias that emerges at demographic intersections by preventing the model from learning discriminative patterns specific to any particular overlapping group, thereby ensuring more equitable treatment across various intersectional groups. Finally, we also develop false positive rate regularizer within the classifier that targets disparities in false positive rates across groups during prediction, addressing outcome bias.

Through extensive experimentation on several real-world datasets spanning healthcare, finance, and image domain, we demonstrate that Diff-Fair consistently outperforms state-of-the-art fairness-aware methods. Our approach achieves substantially higher accuracy (up to 24% improvement on MIMIC datasets) while simultaneously reducing demographic disparities, with demographic parity ratios more than double that of existing methods for intersectional groups. Notably, Diff-Fair significantly minimizes false positive rate differences across sensitive groups, a critical metric in high-stakes domains, while maintaining competitive utility, thereby addressing the fundamental fairness-accuracy trade-off that has challenged prior approaches.

## 2 BACKGROUND

Fairness in machine learning address the concern that automated decision systems may amplify existing societal biases against certain demographic groups Chouldechova & Roth (2020). These systems can inherit biases from historical data, leading to discriminatory outcomes that disproportionately affect marginalized groups. The concept of algorithmic fairness seeks to ensure that machine learning models make decisions that do not systematically disadvantage individuals based on sensitive attributes such as race, gender, age, or disability status.

**Definition 1** (Intersectional Fairness Crenshaw (2013)). *Intersectional fairness considers the overlapping and interdependent systems of discrimination or disadvantage that may affect individuals belonging to multiple sensitive groups. Mathematically, given the sensitive attributes $S = (S_1, S_2, ..., S_m)$, intersectional fairness requires that a model's behavior satisfy the fairness criteria across all possible combinations of attribute values $s \in \mathcal{S}_1 \times \mathcal{S}_2 \times \cdots \times \mathcal{S}_m$, rather than considering each attribute $S_i$ independently.*

**Definition 2** (Demographic Parity Ratio Weerts et al. (2023)). *Demographic parity requires that the probabilities of a positive outcome should be the same across all demographic groups. For intersectional fairness, a classifier $f$ satisfies demographic parity across all intersectional subgroups:*

$$Pr(f(X) = 1|S = s) = Pr(f(X) = 1|S = s'), \forall_{s,s'} \in \mathcal{S} \tag{1}$$

*and their ratio:*

$$DPR = \frac{Pr(f(X) = 1|S = s)}{Pr(f(X) = 1|S = s')}, \forall_{s,s'} \in \mathcal{S} \tag{2}$$

*here, DPR of* 1 *signifies perfect demographic parity Dwork et al. (2012)*

**Definition 3** (False Positive Rate Parity). *False positive rate parity addresses cases where individuals undeserving of a positive outcome are falsely given a positive prediction. It requires that the false positive rates are equal across protected groups:*

$$Pr(f(X) = 1|Y = 0, S = s) = Pr(f(X) = 1|Y = 0, S = s'), \forall_{s,s'} \in \mathcal{S} \tag{3}$$

**Definition 4** (Mutual Information Estimation Kraskov et al. (2004)). *The Mutual information between random variables $X$ and $Y$ is:*

$$I(X;Y) = \int_X \int_Y p(x,y) \log \frac{p(x,y)}{p(x)p(y)} dx dy \tag{4}$$

The mutual information $I(Z;S)$ between a learned representation $Z$ and a sensitive attribute $S$ provides a principled measure of the information leakage about protected characteristics. As we show in Section 3, minimizing this quantity directly promotes demographic parity.

## 3 APPROACH

In this section, we present our proposed framework Diff-Fair, and explain each components. An overview of the proposed architecture can be found in the appendix section Figure 2.

### 3.1 PROBLEM FORMULATION

Given a dataset $\mathcal{D} = \{(x_i, y_i, s_i)\}_{i=1}^N$, where $x_i \in \mathbb{R}^d$ represents features, $y_i \in \{0, 1, \ldots, C-1\}$ denotes the class label with $C$ being the number of classes, and a joint sensitive attribute space $\mathcal{S} = \mathcal{S}_1 \times \mathcal{S}_2 \times \cdots \times \mathcal{S}_m$ (e.g., race $\times$ gender), our goal is to learn representations that simultaneously support accurate multiclass predictions while mitigating bias across intersectional demographic groups.

### 3.2 DIFFUSION MODEL

At the core of our approach is a diffusion model that creates latent representation. Our choice of diffusion models over conventional encoders (e.g. VAEs, standard autoencoders) is motivated by their unique advantages in training mechanism. Unlike VAEs that can suffer from posterior collapse or standard autoencoders that may learn trivial reconstructions, diffusion models are trained to denoise inputs across multiple noise levels, creating representations that are inherently robust to perturbations and capture features Ho et al. (2020).

The forward diffusion process gradually adds Gaussian noise to the data according to a predefined schedule Ho et al. (2020):

$$x_t = \sqrt{\alpha_t} x_0 + \sqrt{1 - \alpha_t} \epsilon, \quad \epsilon \sim \mathcal{N}(0, I) \tag{5}$$

where $t \in \{0, 1, ..., T\}$ represents the diffusion timestep, $\alpha_t \in [0, 1]$ controls noise level, and $x_0$ is the original data point. For the reverse process, we train a neural network $\epsilon_\theta(x, t)$ to predict the noise component allowing us to progressively de-noise and generate samples:

$$\hat{\epsilon} = \epsilon_\theta(x, t) \tag{6}$$

Our diffusion model architecture consists of a time embedding layer that maps discrete timesteps to continuous embeddings $emb(t) \in \mathbb{R}^{d_t}$ and a series of fully connected layers that process the concatenation of noisy data and time embeddings:

$$h_1 = \text{ReLU}([x_t, emb(t)] W_1 + b_1) \tag{7}$$
$$h_2 = \text{ReLU}(h_1 W_2 + b_2) \tag{8}$$
$$\hat{\epsilon} = h_2 W_3 + b_3 \tag{9}$$

Additionally, we leverage the first hidden layer $h_1$ as our encoder to extract latent representations $z = h_1$. This design choice is both theoretically motivated and empirically validated. The first layer receives the noise-corrupted input concatenated with temporal embedding, positioning it to capture essential semantic features immediately after initial denoising. In contrast, deeper layers become

increasingly specialized for the noise prediction task, potentially losing information valuable for downstream classification. Our empirical analysis also supports this design choice. Table 11 in the appendix section shows that representation from $h_1$ significantly outperforming deeper layer representations. Additionally, Table 3 in the appendix sections shows the architecture for handling image datasets.

## 3.3 CLASSIFIER MODULE

A critical component of our framework is the classifier module, which serves dual purpose, e.g. (1) it evaluates the utility of the learned representations for the target prediction task, and (2) it provides the basis for our fairness regularization, particularly for monitoring and mitigating disparate error rates across demographic groups.

### 3.3.1 CLASSIFIER ARCHITECTURE

We implement the classifier as a feed-forward neural network that maps latent representations to class probabilities. Instead of using original input features directly, our classifier operates on the latent space encoded by the diffusion model:

$$h_1 = ReLU(zW_1 + b_1) \tag{10}$$
$$h_2 = ReLU(h_1W_2 + b_2) \tag{11}$$
$$logits = h_2W_3 + b_3 \tag{12}$$
$$p(y|z) = softmax(logits) \tag{13}$$

where $z$ represents the latent representation extracted from the diffusion model. This design choice ensures that our classifier directly evaluates the quality and fairness of the learned representations, rather than relying on the potentially biased features. Table 3 in the appendix sections shows the architecture for handling image datasets.

## 3.4 FAIRNESS AWARE LEARNING

Our approach integrates three complementary fairness mechanisms that collectively address different aspects of algorithmic bias:

### 3.4.1 MUTUAL INFORMATION MINIMIZATION

To ensure that sensitive attributes are not encoded in the latent representations, we use a mutual information estimator based on the Mutual Information Neural Estimator (MINE) technique Belghazi et al. (2018). We train an auxiliary network $T_\phi(z, s)$ that takes a latent representation $z$ and a sensitive attribute $s$ and output a score. The mutual information is then estimated as:

$$\hat{I}(Z; S) = \mathbb{E}_{(z,s)\sim P_{ZS}}[T_\phi(z, s)] - \log \mathbb{E}_{z\sim P_Z, s\sim P_S}[e^{T_\phi(z,s)}] \tag{14}$$

In practice, for a batch of size $B$, we approximate this function as:

$$\mathcal{L}_{\text{MI}} = \hat{I}(Z; S) = \frac{1}{B}\sum_{i=1}^{B} T_\phi(z_i, s_i) - \log\left(\frac{1}{B}\sum_{i=1}^{B} e^{T_\phi(z_i, s_{\pi(i)})}\right) \tag{15}$$

here, $\pi$ is a random permutation of batch indices.

This approach provides not only an estimator but also theoretical guarantees for fairness. Specifically, we can establish the formal relationship between minimizing mutual information and achieving demographic parity.

**Theorem 1.** *Let, $z$ be the latent representation of data $x$ and $s$ be the sensitive attribute, and $f$ be any classifier that depends solely on $z$. If the mutual information $I(Z; S) = 0$, then $f$ satisfies demographic parity:*

$$\Pr(f(z) = y | S = s) = \Pr(f(z) = y) \quad \forall y, s \tag{16}$$

The proof of the theorem can be found in the Appendix section D.

### 3.4.2 INTERSECTIONAL FAIRNESS REGULARIZATION

Unlike many existing approaches that consider sensitive attributes independently, we specifically address intersectional fairness. We further enhance the process for multi-class classification problem. For this, we propose a two-term penalty that, (1) equalizes group centroid to the global mean, and (2) aligns group-specific centroids within each class to their class-conditional mean. Formally, let $\mathcal{G}$ be the set of all intersectional subgroups, then, we define the intersectional regularizer:

$$\mathcal{L}_{\text{intersect}} = \underbrace{\frac{1}{|\mathcal{G}|} \sum_{g \in \mathcal{G}} \left\|\mu_g - \mu\right\|_2^2}_{\text{global subgroup alignment}} + \lambda_{\text{class}} \underbrace{\frac{1}{|\mathcal{G}|} \sum_{c=0}^{C-1} \sum_{g \in \mathcal{G}} \left\|\mu_{g,c} - \mu_c\right\|_2^2}_{\text{class-conditional subgroup alignment}}. \tag{17}$$

where, $\mu_g = \frac{1}{|\{i:s_i=g\}|} \sum_{i:s_i=g} z_i$ is the mean representation for group $g$, $\mu = \frac{1}{N} \sum_{i=1}^{N} z_i$ is the overall population mean, $\mu_{g,c} = \frac{1}{|\{i:s_i=g,y_i=c\}|} \sum_{i:s_i=g,y_i=c} z_i$ is the mean for group $g$ and class $c$ and, $\mu_c = \frac{1}{|\{i:y_i=c\}|} \sum_{i:y_i=c} z_i$ is the mean for class $c$.

**Global subgroup alignment** penalizes any intersectional group's drift away from the population centroid, thereby controlling worst-case deviation across all subgroups.

**Class-conditional subgroup alignment** further enforces that within each predicted label $c$, every subgroup's mean remains close to the class's centroid, guarding against fairness violations inside individual decision boundaries.

By combining these two components, $\mathcal{L}_{\text{intersect}}$ simultaneously mitigates representational bias. The hyperparameter $\lambda_{class}$ enables a data-driven balancing of global versus class-conditional fairness objectives.

### 3.4.3 FALSE POSITIVE RATE REGULARIZATION

To address disparate impact concerns in multi-class settings, we extend the concept of false positive rate (FPR) to handle multiple classes. In binary classification, FPR measures the rate at which negative samples are incorrectly classified as positive. For multiclass settings, we generalize this to measure the rate at which samples from any class are incorrectly classified as belonging to another specific class.

For each class pair $(c, c')$ where $c \neq c'$, and each intersectional group $g$, we define the class specific FPR as:

$$\text{FPR}_{g,c,c'} = \frac{\sum_{i:s_i=g,y_i=c} p(y = c'|x_i)}{|\{i : s_i = g, y_i = c\}|} \tag{18}$$

This measures the rate at which samples from class $c$ are misclassified as class $c'$ for group $g$. Our regularization term penalizes the variance in these rates across groups:

$$\mathcal{L}_{\text{FPR}} = \sum_{c=0}^{C-1} \sum_{c'=0, c' \neq c}^{C-1} \text{Var}(\{\text{FPR}_{g,c,c'}\}_{g \in \mathcal{G}}) \tag{19}$$

This FPR regularization is differentiable and can be directly incorporated into the training process.

### 3.5 JOINT TRAINING PROCESS

We integrate all components into a unified training framework with the following multi-objective loss function:

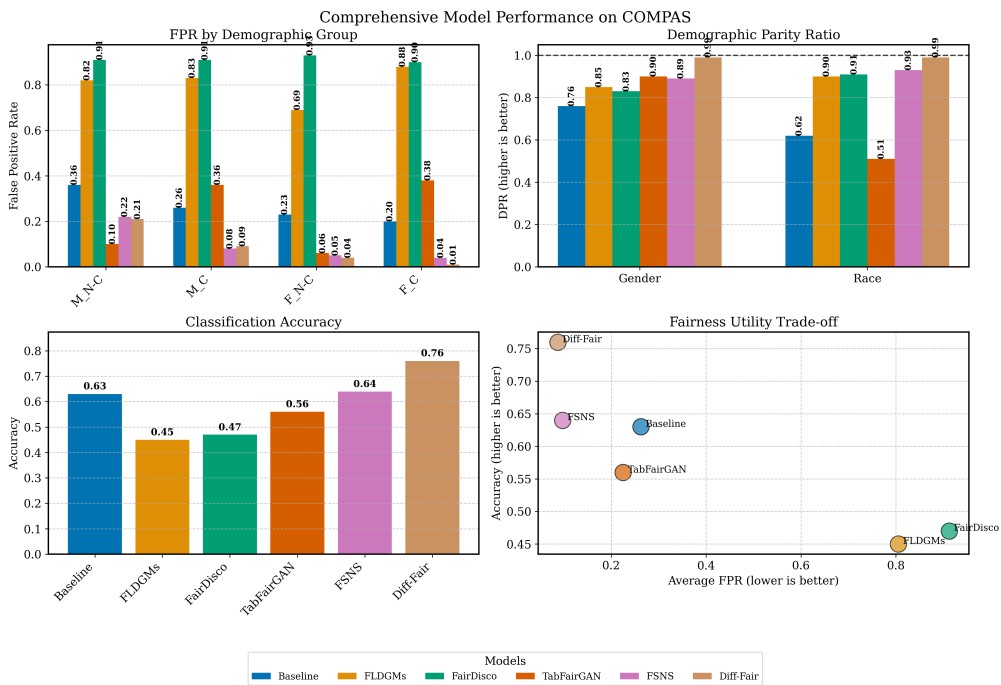

Figure 1: Comprehensive model performance in COMPAS dataset, upper left figure shows the False positive rate (FPR) of intersectional groups, 'M' stands for 'Male' and 'F' stands for 'Female', 'N-C' stands for 'Not-Caucasian' and 'C' stands for 'Caucasian', upper right corner figure shows the Demographic Parity Ratio of models on different sensitive attributes, lower left corner figure shows the accuracy gained by each model and lower right corner shows the Fairness-utility trade off (Accuracy vs Average FPR)

$$\mathcal{L} = \mathcal{L}_{\text{diffusion}} + \lambda_{\text{MI}} \cdot \mathcal{L}_{\text{MI}} + \lambda_{\text{fair}} \cdot \mathcal{L}_{\text{intersect}} + \lambda_{\text{cls}} \cdot \mathcal{L}_{\text{cls}} + \lambda_{\text{FPR}} \cdot \mathcal{L}_{\text{FPR}} \qquad (20)$$

where, $\mathcal{L}_{\text{diffusion}} = ||\hat{\epsilon} - \epsilon||_2^2$ is the diffusion model's noise prediction error, and $\mathcal{L}_{\text{cls}} = -\mathbb{E}_{(z,y)}[log p(y|z)]$ is the standard cross-entropy loss for multiclass classification. Also, $\lambda_{MI}, \lambda_{fair}, \lambda_{cls}, \lambda_{FPR}$ are the hyperparameters that control the trade-off between different objectives. We use `hyperopt` Bergstra et al. (2015), a python library that uses bayesian optimization to search optimal parameters, to find the optimal values for these hyperparameters. Details of this can be found in the appendix section E.

## 4 EXPERIMENTS

In this section, we present a comprehensive evaluation of our proposed approach. We train our framework with six benchmarking datasets and compare against several state-of-the-art models. We split the data in 80/20 for training/testing purpose and also run each experiment five times with different seed values. For baseline comparison, we utilize the authors' original implementation.

### 4.1 DATASETS

To evaluate our proposed framework, we have selected benchmarking datasets spanning multiple application domains and prediction paradigm. Our experimental analysis incorporates financial records (German Credit data), criminal justice information (COMPAS Angwin et al. (2016)), two clinical datasets (MIMIC-III Johnson et al. (2016) and MIMIC-IV Johnson et al. (2023)), and two image datasets (Colored-MNIST Lee et al. (2021) and CelebA Liu et al. (2015) dataset). The healthcare datasets are specifically configured for multi-class prediction tasks, enabling us to assess

the framework's capabilities beyond conventional binary classification scenarios. More details about the dataset can be found in the appendix section C.

## 4.2 BASELINES

To rigorously evaluate our approach's effectiveness, we conduct comprehensive comparisons against state-of-the-art fairness-aware frameworks across multiple application domains. Our comparative analysis includes, adversarial representation learning techniques for intersectional fairness such as FSNS Jang et al. (2024), which implements non-zero-sum adversarial mechanisms to enforce attribute non-separability, TabFairGAN Rajabi & Garibay (2022), which integrates fairness constraints into generative adversarial networks for tabular data synthesis, FairDisco Liu et al. (2022), which operates by minimizing variance correlation between protected and non-protected features, and diffusion-based approaches including FLDGMs Ramachandranpillai et al. (2023), which leverages latent generative mechanisms for fairness-aware synthetic data generation.

## 4.3 EVALUATION METRICS

We evaluate the model in perspective of utility and fairness. For utility assessment, we report standard classification accuracy. Our fairness evaluation incorporates with Demographic Parity ratio (DPR), Equalized Odds ratio (EOR) and False Positive Rate (FPR) across intersectional groups.

## 4.4 RESULTS

We present extensive experimental results across binary classification, multi-class classification for both tabular and image data, demonstrating Diff-Fair's effectiveness in achieving superior fairness-utility trade-offs compared to state-of-the-art methods.

### 4.4.1 BINARY CLASSIFICATION

Figure 1 presents results for the COMPAS dataset, where Diff-Fair achieves 76% accuracy with near-perfect demographic parity ratios (0.99) for both gender and race and score of 0.968 (gender), 0.984 (race) for equalized odds ratios, substantially outperforming baselines. Our method demonstrates superior intersectional fairness, reducing FPRs to 0.04 (non-Caucasian females) and 0.01 (Caucasian females), while maintaining the best utility-fairness trade-off among all models. On German Credit (Table 6), our approach achieves 99% accuracy with zero false positives across almost all intersectional groups which is critical for financial applications. While FairDisco achieves competitive gender fairness (DPR 0.90), Diff-Fair excels in age-group fairness (DPR 0.94) with superior overall utility-fairness balance, establishing strong performance in binary classification that extends to multi-class settings.

### 4.4.2 MULTI-CLASS CLASSIFICATION

We evaluate Diff-Fair on multi-class ICU length-of-stay prediction using MIMIC-III and MIMIC-IV Johnson et al. (2016; 2023). In both datasets, 'Race' has five (MIMIC-III) and eight (MIMIC-IV) categories which grouped with binary gender, this yields 10 and 16 subgroups. We compare our method against three state-of-the-art fair representation learning approaches: FairDisco Liu et al. (2022), FLDGMs Ramachandranpillai et al. (2023), and FSNS Jang et al. (2024). As shown in Figure 3 and Table 7, Diff-Fair attains the best utility and race-parity with accuracy 0.77/0.87 (MIMIC-III/IV), DPR (R) 0.48/0.56 and the lowest average FPR 0.14/0.09. In the equalized odds ratio, our method achieves the strongest score for 'Race' attribute 0.46/0.59. Overall Diff-Fair delivers a superior fairness-utility trade-off through concurrent gains in accuracy, reduced false positive rates, and improved race parity and EOR (R).

### 4.4.3 RESULTS ON IMAGE DATASET

Beside working on tabular data, we expanded our evaluation with image datasets. Table 8 and 9 report accuracy, average false-positive-rate, DPR and EOR. On Colored-MNIST, Diff-Fair attains the best utility and fairness across all metrics with accuracy 0.965, Avg. FPR of 0.003 for color and 0.006 for label outperforming FLDGMs and FairDisco. On CelebA, our method achieves the highest

Table 1: Robustness Analysis

| Noise% | Accuracy | FPR-GAP |
|--------|----------|---------|
| 0 | 0.76 | 0.19 |
| 10 | 0.70 | 0.25 |
| 30 | 0.58 | 0.31 |
| 50 | 0.49 | 0.37 |

Table 2: Ablation Study, Dataset: German Credit

| | DPR(G) | DPR (R) | Acc |
|--------------------|--------|---------|------|
| Full Model | 0.73 | 0.75 | 0.99 |
| w/o MI | 0.45 | 0.56 | 0.98 |
| w/o MI+Fair+FPR | 0.00 | 0.43 | 0.99 |
| w/o Cls+MI+Fair+FPR | 0.00 | 0.43 | 0.28 |

accuracy of 0.98 and the lowest average FPR (0.103). While FairDisco attains higher DPR, Diff-Fair yields the strongest EOR, (0.71/0.67) for hair-color and gender attribute. Overall, Diff-Fair preserves high-predictive performance while improving EOR and maintaining competitive error rates.

### 4.4.4 MODEL ROBUSTNESS

To evaluate our model's resilience to data quality issues, we conducted robustness experiments on the COMPAS dataset by introducing varying levels of label noise. Table 1 presents how performance metrics degrade as noise percentage increases. With clean data (0% noise), our model achieves 76% accuracy with a relatively low FPR-GAP of 0.19. As noise levels increase to 10%, 30%, and 50%, accuracy steadily declines (70%, 58%, and 49% respectively). Notably, fairness degradation follows a similar pattern, with FPR-GAP progressively worsening (0.25, 0.31, and 0.37). These results demonstrate that while noise impacts both accuracy and fairness, Diff-Fair maintains reasonable performance even with moderate corruption ($10 - 30\%$ noise), highlighting its stability in real-world scenarios where data quality is often imperfect.

**Hyperparameter Sensitivity:** We further analyzed Diff-Fair's sensitivity to key hyperparameter choices using COMPAS dataset. Table 12 in the appendix section shows that our framework exhibits notable stability, e.g. $\lambda_{MI}$ variations from 0.001 to 0.1 maintain accuracy between $73 - 76\%$ and strong demographic parity (Avg. DPR of $0.71 - 0.96$). Similarly, $\lambda_{Fair}$ demonstrates robustness with near perfect fairness metrics across lower values. The classification weight $\lambda_{CLS}$ exhibits the expected fairness-utility trade-off, higher values improve accuracy whole slightly reducing fairness.

### 4.4.5 ABLATION STUDY

We conducted an ablation study on the German Credit dataset to evaluate each component's contribution to our framework. Table 2 shows the impact on demographic parity ratios for gender (DPR(G)) and race (DPR(R)), along with classification accuracy. Removing mutual information minimization significantly reduces fairness metrics (DPR(G) drops from 0.73 to 0.45, DPR(R) from 0.75 to 0.56) while maintaining accuracy. When removing multiple fairness components (MI+Fair+FPR), gender fairness collapses completely despite unchanged accuracy. Without the classifier module and fairness components, both utility and fairness severely degrade (Acc = 0.28, DPR (G) = 0.00, DPR (R) = 0.43). These results demonstrate that each component plays a crucial role in our framework, mutual information minimization, fairness regularization prevents protected attribute exploitation, and the classifier maintains utility while respecting fairness constraints.

## 5 DISCUSSION

Our approach to fair representation learning effectively addresses algorithmic bias through a multi-level fairness framework that simultaneously targets representational, group, and outcome fairness. At the representational level, our diffusion-based approach creates latent representations that reduced bias while maintaining high utility ($77 - 87\%$ accuracy on MIMIC datasets, 99% in German Credit). At the group level, our mutual information neural estimator minimizes protected attribute leakage, achieving demographic parity ratios more than double those of comparative methods (0.48 and 0.56 for DPR-Race on MIMIC datasets), with similar improvements on German Credit (0.73 for gender, 0.94 for race). Finally, at the outcome level, our false positive rate regularizer directly addresses disparate impact in model decisions, yielding significantly reduced average FPRs (0.14 and 0.09) on healthcare datasets, perfect FPR (almost all 0.00) in German Credit and more balanced FPRs

across intersectional groups in COMPAS, a crucial improvements for high-stakes domains where algorithmic decisions have significant societal consequences.

Our framework excels particularly in addressing intersectional fairness through our proposed regularizer, which minimizes representational differences across overlapping demographic groups. The COMPAS dataset results (Figure 1) demonstrate this advantage, with our approach effectively mitigating bias at demographic intersections, addressing a fundamental limitation of traditional approaches that may appear fair when evaluated on individual attributes but fail to handle compounded discrimination at their intersections. The practical utility of our approach is further evidenced by its robustness to data quality issues, maintaining acceptable performance even with moderate label noise (10-30%), vital for real-world deployment scenarios with imperfect data.

**Limitations and Future Work**   Despite the promising results of our framework, several challenges remain. First, our approach requires careful hyperparameter tuning to balance fairness constraints with model utility, as evidenced by the dataset-specific configurations in our experimental setup (appendix Table 4). Future work could explore automated methods for determining optimal fairness-utility trade-offs based on domain-specific requirements.

**Societal Impact**   The societal implications of fair representation learning extend beyond technical metrics. By developing models that specifically targets for intersectional fairness concerns, our work contributes to more equitable algorithmic decision-making across diverse applications. In high-stakes domains like criminal justice, lending, or healthcare, biased representations can reinforce and amplify existing social inequities. For example, recidivism prediction algorithms that produce higher false positive rates for certain demographic groups can cause disadvantage. Our approach, with its explicit focus on FPR parity and intersectional fairness, directly addresses these concerns. However, we acknowledge no technical solution alone can address the structural inequalities embedded in our social systems. Fair representation learning should be viewed as one component of a broader socio-technical approach to fairness that includes diverse teams, external audits, and community involvement.

## 6   CONCLUSION

We have presented Diff-Fair, a novel framework that leverages diffusion models to address algorithmic bias through fair representation learning. Our approach makes several key contributions to fairness research. First, it demonstrates that diffusion models, typically employed for generative tasks, can be effectively repurposed for learning fair representations that maintain high utility. Second, it integrates complementary mechanisms that simultaneously address fairness at the representational, group, and outcome levels, creating a comprehensive solution to algorithmic bias. Third, it specifically targets intersectional fairness through a dedicated regularizer that prevents models from learning discriminative patterns specific to overlapping demographic groups. Through extensive experimentation across diverse domains including criminal justice, finance, and healthcare, we demonstrate that Diff-Fair consistently outperforms state-of-the-art approaches in reducing demographic disparities and false positive rate differences while maintaining competitive accuracy for both binary and multi-class classification tasks. The framework's robustness to moderate data quality issues further supports its potential for real-world applications. As AI systems continue to influence critical decisions across society, our work offers a promising path toward more equitable outcomes for all individuals, regardless their demographic characteristics. By addressing the complex challenge of intersectional fairness while maintaining strong predictive performance, Diff-Fair represents an important step in building AI systems that can help reduce existing societal disparities. We believe this approach opens new possibilities for fair representation learning across a wide range of high-stakes applications, ultimately contributing to the development of more trustworthy and beneficial AI technologies that can serve diverse populations equitably.

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

## A    APPENDIX

## B    RELATED WORKS

### B.1    FAIRNESS IN MACHINE LEARNING

The development of methods to extract unbiased representations from inherently biased datasets has gained significant traction in the machine learning community. Zemel et al. pioneered this area with Learning Fair Representations (LFR), which conceptualizes the challenge as an optimization problem where learned representations must support accurate predictions while maintaining fairness metrics Zemel et al. (2013). Taking a different approach, FairDisco focuses on specifically reducing the correlation variance between protected and non-protected attributes to generate representations free from systematic bias Liu et al. (2022). Simultaneously, adversarial frameworks have become increasingly prevalent, where dedicated adversarial components work to identify potential bias patterns, thereby enforcing equity constraints through competitive optimization Gao et al. (2022); Madras et al. (2018); Xu et al. (2021). Complementing these representation learning techniques, several fair generative approaches have been developed to create synthetic data that maintains utility while diminishing inherent biases. TabFairGAN Rajabi & Garibay (2022) exemplifies this direction by embedding fairness objectives directly into GAN architectures for tabular data generation. Despite their promising performance regarding both fairness metrics and predictive accuracy, many of these approaches exhibit limitations, they often address only single protected attributes and sometimes are restricted to binary classification scenarios Rajabi & Garibay (2022); Liu et al. (2022); Ramachandranpillai et al. (2023).

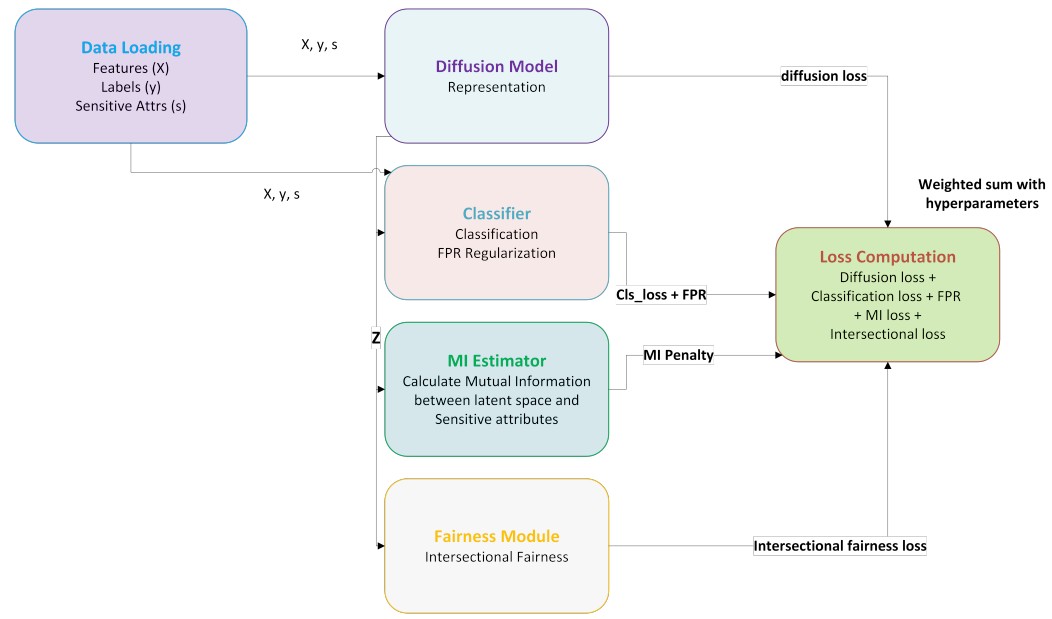

Figure 2: Diff-Fair: Our Proposed Architecture

### B.2 INTERSECTIONAL FAIRNESS APPROACH

Recent advances in fairness research have expanded beyond single-attribute considerations to address intersectional fairness, which examines how multiple protected characteristics interact to create unique patterns of bias. This intersectional perspective recognizes that fairness interventions focusing solely on individual attributes like race or gender in isolation may fail to address compounded discrimination experienced at their intersections. Several approaches have been proposed in this domain, including FSNS Jang et al. (2024), which pioneered techniques for measuring fairness across overlapping demographic subgroups. As FSNS operates on standard tabular, we have included it as a baseline in our experimental comparisons. In contrast, other methods such as RELIANT Dong et al. (2023) and FairGKD Zhu et al. (2024) leverage knowledge distillation techniques alongside graph-structured constraints to mitigate intersectional bias. However, these graph-based approaches are fundamentally limited to contexts where relationships between entities are specifically modeled as edges in a graph. This domain-specific constraint places them outside the scope of comparison for our work, which focuses on tabular data without inherent graph structure. The limitations of existing methods highlight the need for more versatile approaches to intersectional fairness that can effectively operate across various data modalities, particularly for structured tabular data common in high-stakes decision-making systems.

## C DATASETS

Our evaluation framework was tested on six benchmark datasets commonly used in fairness research:

1. **COMPAS Recidivism:** Comprising criminal records, demographic information, and risk assessments for approximately 7,000 individuals, this dataset's target variable indicates whether a person reoffended within a two-year timeframe. Our analysis considers both gender and race as protected characteristics.

2. **German Credit:** This financial dataset consists of 1,000 records with 20 features describing applicant profiles for credit risk assessment. The protected attributes in our experiments include age-group and gender.

3. **MIMIC-III Clinical Data:** This authorized-access healthcare dataset encompasses over 60,000 intensive care unit admissions, including vital measurements, medication records, and

diagnostic information. Our prediction task classifies patients into four ordered categories based on hospital stay duration: under 3 days, 3-7 days, 7-14 days, and beyond 14 days. We examine fairness across gender and five racial categories, creating 10 distinct intersectional subgroups. This configuration enables us to evaluate fairness in multi-class prediction scenarios across diverse demographic intersections in critical healthcare applications.

4. **MIMIC-IV Clinical Data:** As an expanded version of MIMIC-III, this authorized-access dataset contains 94,444 unique ICU admissions. We maintain the same length-of-stay prediction task as with the MIMIC-III dataset for consistency in our healthcare domain experiments.

5. **Colored-MNIST:** We conducted experiments using Colored-MNIST data as described in Liu et al. (2022); Lee et al. (2021). This is an image dataset and here color serves as the sensitive attribute while being uncorrelated with the actual digit content. We manipulate color intensity to create digits with varying shades—from faint to vivid—within a single hue (red, green, or blue). We use the color of the digits and digit labels as sensitive attributes. As we have three colors and 10 digit labels, so the intersectional group would be (red_label0, red_label1,...etc).

6. **CelebA:** For our experiment, we used another image dataset titled CelebA Liu et al. (2015). This dataset contains more than 200K celebrity photos with 40 attribute annotations. We used "hair-color" and "gender" as sensitive attribute. So the intersectional group would be (hair_color0_male, hair_color1_male,..,etc).

## D   THEORY AND PROOF

**Theorem 2.** *Let, $z$ be the latent representation of data $x$ and $s$ be the sensitive attribute, and $f$ be any classifier that depends solely on $z$. If the mutual information $I(Z; S) = 0$, then $f$ satisfies demographic parity:*

$$\Pr(f(z) = y | S = s) = \Pr(f(z) = y) \quad \forall y, s \tag{21}$$

**Proof**   When $I(Z; S) = 0$, the joint probability distribution factorizes as $p(z, s) = p(z)p(s)$ almost everywhere Cover (1999). For any classifier $f$ that depends solely on $Z$ through some conditional distribution $p(f(z) = y | z)$, we have:

$$\Pr(f(z) = y, S = s) = \int p(f(z) = y | z)p(z, s)dz \tag{22}$$

$$= \int p(f(z) = y | z)p(z)p(s)dz \tag{23}$$

$$= p(s) \int p(f(z) = y | z)p(z)dz \tag{24}$$

$$= p(s)\Pr(f(z) = y) \tag{25}$$

Dividing both sides by $p(s) = \Pr(S = s)$ yields the demographic parity criterion Dwork et al. (2012); Feldman et al. (2015):

$$\Pr(f(z) = y | S = s) = \Pr(f(z) = y) \tag{26}$$

## E   HYPERPARAMETERS

Our framework is consists of several hyperparameters to aid the various loss mechanism to achieve final objectives. To find the optimal values for these hyperparameters, we use `Hyperopt` Bergstra et al. (2015). The objective function on the Hyperopt was to penalize maximum FPR-variance and FPR-difference. We use 10 trials for each dataset (except for colored-MNIST and CelebA due to their huge memory footprint) and achieve the parameters presented in Table 4.

## F   DETAILED RESULTS

Along with the Figure 1 and 3, we also present the tabular values of the evaluation metrics. Table 5 records the evaluation metrics of COMPAS dataset, Table 6 represents for the German Credit and

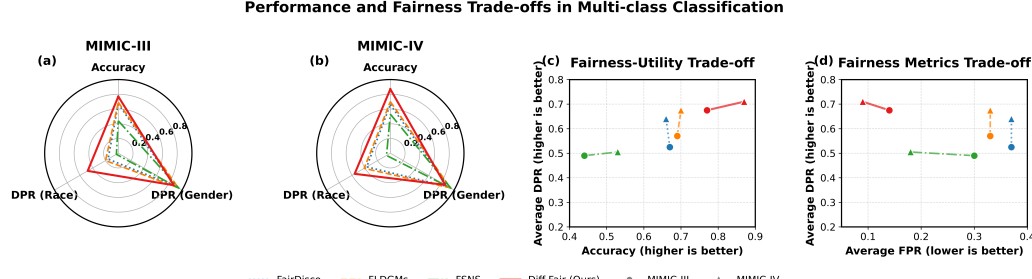

Figure 3: Performance of Diff-Fair in Multi-class Dataset

Table 7 presents the MIMIC-III, IV's experimental results. Table 8 and 9 report the results for image datasets.

Table 3: Diff-Fair Model Architecture (shared across datasets)

| *Training Settings* | |
|---|---|
| Learning Rate | $1 \times 10^{-4}$ |
| Batch Size | 256 |
| Epochs | 1000 |
| Diffusion Steps | 1000 |
| Noise Scale $\alpha$ | 0.9 |

| **Component** | **Tabular Data** | **Image Data** |
|---|---|---|
| **Diffusion Model** | | |
| Input Dim | Dataset-specific | $3 \times 64 \times 64$ |
| Hidden Dim | 64 | 256 |
| Time Emb. Dim | 16 | 32 |
| Architecture | MLP | CNN |
| Layers | $[x + t] \rightarrow 64 \rightarrow 64 \rightarrow x$ | Conv layers with pooling |
| **MI Estimator** | | |
| Latent Dim | 64 | $128 \times 16 \times 16$ |
| Hidden Dim | 64 | 64 |
| Architecture | MLP | CNN + Global Pool |
| Output | 1 | 1 |
| **Classifier** | | |
| Input Dim | Latent space | Latent space |
| Hidden Dim | 64 | 64 |
| Architecture | MLP | CNN + Global Pool |
| Output | # classes | # classes |

Table 4: Loss weight configurations per dataset

| **Dataset** | $\lambda_{\text{cls}}$ | $\lambda_{\text{MI}}$ | $\lambda_{\text{fair}}$ | $\lambda_{\text{FPR}}$ |
|---|---|---|---|---|
| COMPAS | 0.50 | 0.10 | 0.10 | 0.10 |
| German Credit | 1.67 | 0.01 | 0.01 | 0.02 |
| MIMIC-III | 1.60 | 0.24 | 0.14 | 0.18 |
| MIMIC-IV | 0.96 | 0.16 | 0.23 | 0.27 |
| Colored-MNIST | 1.0 | 0.05 | 0.05 | 0.05 |
| CelebA | 1.0 | 0.01 | 0.01 | 0.01 |

Table 5: COMPAS Dataset (bold indicates best result, ↑ indicates higher as better, ↓ indicates lower as better)

| Gender | Race | FLDGMs FPR | FairDisco FPR | TabFairGAN FPR | FSNS FPR | Baseline FPR | Ours FPR |
|---|---|---|---|---|---|---|---|
| Male | Not-Caucasian(↓) | $0.81 \pm 0.01$ | $0.90 \pm 0.01$ | $\mathbf{0.10} \pm 0.02$ | $0.21 \pm 0.01$ | $0.35 \pm 0.01$ | $0.21 \pm 0.013$ |
| Male | Caucasian(↓) | $0.82 \pm 0.01$ | $0.90 \pm 0.03$ | $0.35 \pm 0.01$ | $\mathbf{0.08} \pm 0.01$ | $0.26 \pm 0.01$ | $0.09 \pm 0.013$ |
| Female | Not-Caucasian(↓) | $0.69 \pm 0.01$ | $0.93 \pm 0.01$ | $0.06 \pm 0.01$ | $0.05 \pm 0.01$ | $0.23 \pm 0.01$ | $\mathbf{0.04} \pm 0.014$ |
| Female | Caucasian(↓) | $0.88 \pm 0.01$ | $0.90 \pm 0.01$ | $0.38 \pm 0.01$ | $0.04 \pm 0.01$ | $0.20 \pm 0.01$ | $\mathbf{0.01} \pm 0.004$ |
| | Accuracy(↑) | $0.45 \pm 0.01$ | $0.47 \pm 0.01$ | $0.56 \pm 0.01$ | $0.64 \pm 0.01$ | $0.63 \pm 0.01$ | $\mathbf{0.758} \pm 0.004$ |
| DPR | Gender(↑) | $0.85 \pm 0.01$ | $0.83 \pm 0.01$ | $0.90 \pm 0.01$ | $0.89 \pm 0.01$ | $0.76 \pm 0.01$ | $\mathbf{0.99} \pm 0.01$ |
| DPR | Race(↑) | $0.90 \pm 0.01$ | $0.91 \pm 0.01$ | $0.51 \pm 0.01$ | $0.93 \pm 0.01$ | $0.62 \pm 0.01$ | $\mathbf{0.99} \pm 0.01$ |
| EOR | Gender(↑) | $0.85 \pm 0.01$ | $0.83 \pm 0.01$ | $0.90 \pm 0.01$ | $0.89 \pm 0.01$ | $0.76 \pm 0.01$ | $\mathbf{0.968} \pm 0.013$ |
| EOR | Race(↑) | $0.90 \pm 0.01$ | $0.91 \pm 0.01$ | $0.51 \pm 0.01$ | $0.93 \pm 0.01$ | $0.62 \pm 0.01$ | $\mathbf{0.984} \pm 0.008$ |

Table 6: German Credit Dataset (bold indicates best result, ↑ indicates higher as better, ↓ indicates lower as better)

| Gender | Age-group | FLDGMs FPR | FairDisco FPR | FSNS FPR | Baseline FPR | Ours FPR |
|---|---|---|---|---|---|---|
| Female | Young(↓) | $0.33 \pm 0.01$ | $0.25 \pm 0.01$ | $0.25 \pm 0.01$ | $0.58 \pm 0.01$ | $\mathbf{0.00} \pm 0.00$ |
| Female | Old(↓) | $0.29 \pm 0.01$ | $0.33 \pm 0.01$ | $0.10 \pm 0.01$ | $0.23 \pm 0.01$ | $\mathbf{0.01} \pm 0.01$ |
| Male | Young(↓) | $0.00 \pm 0.01$ | $0.28 \pm 0.01$ | $0.16 \pm 0.01$ | $0.71 \pm 0.01$ | $\mathbf{0.00} \pm 0.00$ |
| Male | Old(↓) | $0.25 \pm 0.01$ | $0.23 \pm 0.01$ | $0.12 \pm 0.01$ | $0.26 \pm 0.01$ | $\mathbf{0.00} \pm 0.00$ |
| | Accuracy(↑) | $0.63 \pm 0.01$ | $0.61 \pm 0.01$ | $0.76 \pm 0.01$ | $0.57 \pm 0.01$ | $\mathbf{0.99} \pm 0.01$ |
| DPR | Gender(↑) | $0.89 \pm 0.01$ | $\mathbf{0.90} \pm 0.01$ | $0.69 \pm 0.01$ | $0.63 \pm 0.01$ | $0.73 \pm 0.01$ |
| DPR | Age-group(↑) | $0.75 \pm 0.01$ | $0.69 \pm 0.01$ | $0.78 \pm 0.01$ | $0.76 \pm 0.01$ | $\mathbf{0.94} \pm 0.01$ |
| EOR | Gender(↑) | $\mathbf{0.83} \pm 0.01$ | $0.81 \pm 0.01$ | $0.71 \pm 0.01$ | $0.61 \pm 0.01$ | $0.71 \pm 0.01$ |
| EOR | Age-group(↑) | $0.23 \pm 0.01$ | $0.47 \pm 0.01$ | $0.67 \pm 0.01$ | $0.78 \pm 0.01$ | $\mathbf{0.90} \pm 0.01$ |

Table 7: Multi-class Classification on MIMIC-III and MIMIC-IV dataset, Here 'G' represents Gender and 'R' represents Race, (bold indicates best result, ↑ indicates higher as better, ↓ indicates lower as better)

| Dataset | Evaluation | FairDisco | FLDGMs | FSNS | Ours |
|---|---|---|---|---|---|
| MIMIC-III | Accuracy(↑) | $0.67 \pm 0.01$ | $0.69 \pm 0.01$ | $0.44 \pm 0.01$ | $\mathbf{0.77} \pm 0.01$ |
| | DPR (G)(↑) | $0.89 \pm 0.01$ | $0.94 \pm 0.01$ | $\mathbf{0.95} \pm 0.01$ | $0.87 \pm 0.01$ |
| | DPR (R)(↑) | $0.16 \pm 0.01$ | $0.20 \pm 0.01$ | $0.03 \pm 0.01$ | $\mathbf{0.48} \pm 0.01$ |
| | EOR (G)(↑) | $0.86 \pm 0.01$ | $0.89 \pm 0.01$ | $\mathbf{0.91} \pm 0.01$ | $0.83 \pm 0.01$ |
| | EOR (R)(↑) | $0.11 \pm 0.01$ | $0.16 \pm 0.01$ | $0.05 \pm 0.01$ | $\mathbf{0.46} \pm 0.01$ |
| | Avg FPR(↓) | $0.37 \pm 0.01$ | $0.33 \pm 0.01$ | $0.30 \pm 0.01$ | $\mathbf{0.14} \pm 0.01$ |
| MIMIC-IV | Accuracy(↑) | $0.66 \pm 0.01$ | $0.70 \pm 0.01$ | $0.53 \pm 0.01$ | $\mathbf{0.87} \pm 0.01$ |
| | DPR (G)(↑) | $0.91 \pm 0.01$ | $0.94 \pm 0.01$ | $\mathbf{0.95} \pm 0.01$ | $0.86 \pm 0.01$ |
| | DPR (R)(↑) | $0.37 \pm 0.01$ | $0.41 \pm 0.01$ | $0.06 \pm 0.01$ | $\mathbf{0.56} \pm 0.01$ |
| | EOR (G)(↑) | $0.89 \pm 0.01$ | $0.91 \pm 0.01$ | $\mathbf{0.93} \pm 0.01$ | $0.88 \pm 0.01$ |
| | EOR (R)(↑) | $0.18 \pm 0.01$ | $0.21 \pm 0.01$ | $0.02 \pm 0.01$ | $\mathbf{0.59} \pm 0.01$ |
| | Avg FPR(↓) | $0.37 \pm 0.01$ | $0.33 \pm 0.01$ | $0.18 \pm 0.01$ | $\mathbf{0.09} \pm 0.01$ |

Table 8: Colored-MNIST Dataset (bold indicates best result, ↑ indicates higher as better, ↓ indicates lower as better)

|  |  | FLDGMs | FairDisco | Ours |
|---|---|---|---|---|
| Color | Avg. FPR (↓) | $0.23 \pm 0.01$ | $0.17 \pm 0.01$ | $\mathbf{0.003} \pm 0.01$ |
| Label | Avg. FPR (↓) | $0.25 \pm 0.01$ | $0.14 \pm 0.01$ | $\mathbf{0.006} \pm 0.01$ |
|  | Accuracy (↑) | $0.53 \pm 0.01$ | $0.57 \pm 0.01$ | $\mathbf{0.965} \pm 0.01$ |
| DPR | Color(↑) | $0.73 \pm 0.01$ | $0.81 \pm 0.01$ | $\mathbf{0.89} \pm 0.01$ |
| DPR | Label(↑) | $0.49 \pm 0.01$ | $0.57 \pm 0.01$ | $\mathbf{0.97} \pm 0.01$ |
| EOR | Color(↑) | $0.11 \pm 0.01$ | $0.31 \pm 0.01$ | $\mathbf{0.86} \pm 0.01$ |
| EOR | Label(↑) | $0.07 \pm 0.01$ | $0.23 \pm 0.01$ | $\mathbf{0.89} \pm 0.01$ |

Table 9: CelebA Dataset (bold indicates best result, ↑ indicates higher as better, ↓ indicates lower as better)

|  |  | FLDGMs | FairDisco | Ours |
|---|---|---|---|---|
| Hair-color | Avg. FPR (↓) | $0.33 \pm 0.01$ | $0.27 \pm 0.01$ | $\mathbf{0.103} \pm 0.01$ |
| Gender | Avg. FPR (↓) | $0.18 \pm 0.01$ | $\mathbf{0.13} \pm 0.01$ | $0.145 \pm 0.01$ |
|  | Accuracy (↑) | $0.75 \pm 0.01$ | $0.71 \pm 0.01$ | $\mathbf{0.98} \pm 0.012$ |
| DPR | Hair-color(↑) | $0.81 \pm 0.01$ | $\mathbf{0.84} \pm 0.01$ | $0.76 \pm 0.01$ |
| DPR | Gender(↑) | $0.73 \pm 0.01$ | $\mathbf{0.76} \pm 0.01$ | $0.63 \pm 0.01$ |
| EOR | Hair-color(↑) | $0.26 \pm 0.01$ | $0.34 \pm 0.01$ | $\mathbf{0.71} \pm 0.01$ |
| EOR | Gender(↑) | $0.18 \pm 0.01$ | $0.35 \pm 0.01$ | $\mathbf{0.67} \pm 0.01$ |

## G   COMPUTATIONAL EFFICIENCY

We analyze the computational requirements of Diff-Fair across datasets of varying complexity to assess its practical applicability. As shown in Table 10, our framework demonstrates reasonable resource demands even for complex multi-class prediction tasks. The smaller datasets (German Credit, COMPAS) are processed efficiently with minimal resource requirements (346-378 MB of GPU memory) and rapid training times (1-4 minutes per run). For the larger clinical datasets involving multi-class prediction across numerous intersectional groups, Diff-Fair requires moderately increased but still practical computational resources (762-835 MB) with training times of 71-84 minutes. However, image dataset took a while to complete the training. We ran all the experiment in a workstation consists of a single NVIDIA GeForce RTX 3090 Ti GPU with 24GB of VRAM, AMD Ryzen 9 5900x 12-core processor, 128 GB RAM. We also present Algorithm 1 of our framework training process.

Table 10: Memory Usage and Time

| Dataset | Time/run | Memory usage | Epochs | Batch Size |
|---|---|---|---|---|
| COMPAS | 3 min 42 sec | 378 MB | 1000 | 256 |
| German Credit | 1 min | 346 MB | 1000 | 256 |
| MIMIC-III | 84 min 14 sec | 835 MB | 1000 | 256 |
| MIMIC-IV | 71 min 05 sec | 762 MB | 1000 | 256 |
| Colored-MNIST | 41 min 53 sec | 869 MB | 1000 | 256 |
| CelebA | 20 hours 47 min | 6789 MB | 1000 | 256 |

Table 11: Hidden Layer Timing

| Layer | Acc | Avg. FPR | Avg. DPR | Time |
|---|---|---|---|---|
| $h1$ (our approach) | 0.758 | 0.09 | 0.98 | 3 min 42 sec |
| $h2$ | 0.53 | 0.49 | 0.84 | 4 min 49 sec |

Table 12: Sensitivity Analysis of Hyperparameters, Dataset: COMPAS

| $\lambda_{MI}$ | Acc | Avg. FPR | Avg. DPR |
|---|---|---|---|
| 0.001 | 0.73 | 0.10 | 0.88 |
| 0.01 | 0.76 | 0.09 | 0.96 |
| 0.1 (optimal) | 0.758 | 0.09 | 0.71 |
| 0.5 | 0.72 | 0.14 | 0.71 |
| $\lambda_{Fair}$ | | | |
| 0.001 | 0.75 | 0.1032 | 0.99 |
| 0.01 | 0.73 | 0.1013 | 0.99 |
| 0.1 (optimal) | 0.758 | 0.09 | 0.98 |
| 0.5 | 0.76 | 0.09 | 0.68 |
| $\lambda_{CLS}$ | | | |
| 0.1 | 0.71 | 0.12 | 0.70 |
| 0.5 (optimal) | 0.758 | 0.09 | 0.98 |
| 1.0 | 0.78 | 0.14 | 0.95 |
| 2.0 | 0.80 | 0.11 | 0.91 |

---

**Algorithm 1** Diff-Fair: Diffusion-based Fair Representation Learning

---

**Require:** Dataset $\mathcal{D} = \{(x_i, y_i, s_i)\}_{i=1}^{N}$, hyperparameters $\lambda_{MI}, \lambda_{fair}, \lambda_{cls}, \lambda_{FPR}$
**Ensure:** Trained diffusion model $\epsilon_\theta$, classifier $f$

1: Initialize diffusion model $\epsilon_\theta$, MI estimator $T_\phi$, and classifier $f$
2: **for** epoch = 1 to epochs **do**
3:    **for** each batch $(x, y, s)$ in $\mathcal{D}$ **do**
4:       $t \sim \text{Uniform}(0, T)$ {Sample diffusion timestep}
5:       $\epsilon \sim \mathcal{N}(0, I)$ {Sample noise}
6:       $x_t = \sqrt{\alpha_t}x + \sqrt{1 - \alpha_t}\epsilon$ {Add noise to data}
7:       $\hat{\epsilon} = \epsilon_\theta(x_t, t)$ {Predict noise}
8:       $L_{diffusion} = \|\hat{\epsilon} - \epsilon\|_2^2$ {Diffusion loss}
9:       $z = \text{Encoder}(x_t, t)$ {Extract latent representations}
10:      $\hat{y} = f(z)$ {Predict labels from latents}
11:      $L_{cls} = \text{CrossEntropy}(\hat{y}, y)$ {Classification loss}
12:      $L_{MI} = \mathbb{E}_{(z,s)}[T_\phi(z, s)] - \log(\mathbb{E}_{z, s_{\text{shuffled}}}[e^{T_\phi(z, s_{\text{shuffled}})}])$ {MI loss}
13:      $L_{intersect} = \frac{1}{|G|}\sum_{g \in G}\|\mu_g - \mu\|_2^2 + \lambda_{class}\frac{1}{|G|}\sum_{c=0}^{C-1}\sum_{g \in G}\|\mu_{g,c} - \mu_c\|_2^2$ {Intersectional fairness}
14:      $L_{FPR} = \sum_{c=0}^{C-1}\sum_{c' \neq c}\text{Var}(\{\text{FPR}_{g,c,c'}\}_{g \in G})$ {FPR regularization}
15:      $L = L_{diffusion} + \lambda_{MI} \cdot L_{MI} + \lambda_{fair} \cdot L_{intersect} + \lambda_{cls} \cdot L_{cls} + \lambda_{FPR} \cdot L_{FPR}$ {Total loss}
16:      Update parameters of $\epsilon_\theta$, $T_\phi$, and $f$ using gradients from $L$
17:    **end for**
18: **end for**
19: **return** $\epsilon_\theta, f$

---

