# OpenReview forum: "Diff-Fair: Mitigating Intersectional Bias Through Diffusion-Driven Fair Representation"
_ICLR.cc/2026/Conference — Submitted to ICLR 2026_

### Official Review · Reviewer_LwKH · 2025-10-21

**Soundness:** 2
**Presentation:** 2
**Contribution:** 2
**Rating:** 2
**Confidence:** 3

**Summary:**

The paper proposes Diff-Fair, a diffusion-based framework for fair representation learning under intersectional sensitive attributes. The proposed method uses a diffusion encoder to produce latent features, trains a classifier on those features, and adds three components for fairness: a mutual-information term to limit leakage of sensitive attributes, an intersectional regularizer that aligns subgroup centroids globally and within classes, and a false-positive-rate regularizer that discourages disparities across groups. Experiments on tabular, clinical, and image datasets report higher accuracy with improved demographic parity, equalized odds, and lower false-positive gaps compared to several baselines.

**Strengths:**

1. This paper studies a timely problem focused on intersectional fairness with a unified representation-level and outcome-level treatment. The write-up is accessible and modular, which helps readers map each mechanism to a specific failure mode.

2. The paper includes a broad empirical scope across six datasets spanning tabular, clinical, and image domains, plus reporting of per-group metrics and some sensitivity studies.

3. There is a clear engineering of a practical training recipe that combines diffusion representations with add-on fairness terms and a classifier, together with a basic runtime and memory table.

**Weaknesses:**

1. Incomplete ablations. Provide clean component-level ablations across datasets for the mutual-information term, centroid alignment, and FPR regularizer, plus negative controls such as random or mismatched subgroup labels.

2. Statistical and experimental rigor could be stronger where claims are large. Although the tables report a mean and a small standard deviation, there is no confidence interval, no statistical test across seeds, and no report of sensitivity to the number of intersectional groups or to extreme imbalance. For clinical datasets, the reported accuracy gains are large, yet there is limited analysis of class imbalance, prevalence shift, or calibration.

3. There is limited exploration of how results change with more groups, different subgroup granularity, or other encoders.

4. Baselines are missing. The paper covers several fair-representation methods, but there are missing controls that would isolate where the gains come from. For example, a reduction-based post-processing method, a strong reweighting baseline, a recent alignment baseline that handles many subgroup combinations, and a non-diffusion encoder with the same three fairness terms.

**Questions:**

1. If you replace the diffusion encoder with a non-diffusion encoder that has comparable capacity and keep the three fairness terms unchanged, how much of the gain remains?

2. Can you provide per-dataset ablations that remove one component at a time, mutual-information term only, centroid alignment only, and  FPR term only to show each term’s marginal contribution on COMPAS, German Credit, and both MIMIC datasets?

3. Do subgroup-centroid alignments reduce between-class margins or increase specific confusions?

4. How do performance and fairness change as the number of intersectional groups increases or when small groups become very imbalanced? Any evidence on the stability of the regularizer under these settings?

---

### Official Review · Reviewer_F8hK · 2025-10-30

**Soundness:** 2
**Presentation:** 3
**Contribution:** 2
**Rating:** 2
**Confidence:** 4

**Summary:**

Diff-Fair is a diffusion-based fair-representation framework that jointly enforces demographic parity (via mutual-information minimization), intersectional alignment (centroid regularization), and false-positive-rate parity (via a multiclass FPR-variance penalty), improving accuracy–fairness trade-offs over recent baselines on tabular and image benchmarks.

**Strengths:**

The paper addresses the important and challenging problem of intersectional fairness and explores a novel use of diffusion models for fair representation learning. While the individual fairness components are known, their integration within a diffusion framework is an interesting and reasonably well-motivated direction. The paper is clearly written and provides empirical evidence across both tabular and image domains, showing improved fairness metrics while maintaining competitive accuracy compared to the presented existing methods.

**Weaknesses:**

- The contribution is primarily integration-focused—known fairness regularizers are combined within a diffusion framework—without clear evidence that diffusion (vs. simpler encoders) is essential. The improvement might stem from the regularizers rather than the diffusion mechanism itself. There’s no strong ablation comparing diffusion vs. simpler encoders (e.g., autoencoder or MLP) trained under identical fairness losses.
- The German Credit results appear unreliable. Multiple configurations report ~99% accuracy (Tables 2 and 6) with near-zero FPRs (Table 6), which is not typical for this dataset. Prior benchmarks generally achieve only ~70–83% accuracy, so such performance suggests possible data leakage or a non-standard split;  rather than a real performance gain [1, 2].

[1] Zemel, Rich, et al. "Learning fair representations." International conference on machine learning. PMLR, 2013.

[2] Liao, Yiqiao, and Parinaz Naghizadeh. "Social bias meets data bias: The impacts of labeling and measurement errors on fairness criteria." Proceedings of the AAAI Conference on Artificial Intelligence. Vol. 37. No. 7. 2023.

- The paper does not discuss the theoretical incompatibility [3] between demographic parity and error-rate–based criteria (equal opportunity/equalized odds) when base rates differ, raising questions about the coherence of the joint objective.

[3] Jon M. Kleinberg, Sendhil Mullainathan, Manish Raghavan:
Inherent Trade-Offs in the Fair Determination of Risk Scores. ITCS 2017: 43:1-43:23

- Although the paper reports Equalized Odds (EOR) in its evaluation, the objective optimizes only an across-group FPR-variance regularizer and the total loss contains no term that enforces EO/TPR parity

**Questions:**

1) How does the diffusion backbone concretely contribute to fairness beyond acting as a generic encoder? Have you run AE/MLP ablations with comparable fairness results to isolate the accuracy gain from the diffusion backbone?

2)  Could the authors clarify how the model achieves ~99% accuracy with near-zero FPRs on German Credit?—far above the ~70–83% typically reported—and what checks did you perform to rule out data leakage or an incorrect train/test split?
 -The model labeled “Baseline” in Tables 5–6 is not defined. Could the authors clarify what this Baseline corresponds to (e.g., unaware classifier, architecture, and training setup)? If it is the unaware classifier, it would also be helpful to explain why Diff-Fair achieves higher accuracy than Baseline(e.g., ~0.57 vs. ~0.99), since fairness regularization often introduces trade-offs rather than accuracy gains.
3) Given the known incompatibility between demographic parity and equal opportunity, how do you reconcile enforcing both objectives simultaneously?

---

### Official Review · Reviewer_6QB7 · 2025-10-30

**Soundness:** 2
**Presentation:** 3
**Contribution:** 2
**Rating:** 4
**Confidence:** 4

**Summary:**

This paper proposes Diff-Fair, a novel framework aimed at mitigating intersectional bias in algorithms through diffusion-driven representation learning. The core of the method is the use of a diffusion model as an encoder, which is jointly trained with a multi-objective loss function comprising five components:
* The diffusion model's own denoising loss ($\mathcal{L}_{diffusion}$)
* A mutual information loss ($\mathcal{L}_{MI}$) to minimize sensitive information leakage
* A newly proposed intersectional fairness regularizer ($\mathcal{L}_{intersect}$) to align subgroup representations
* A newly proposed false positive rate regularizer ($\mathcal{L}_{FPR}$) to equalize error rates in multi-class settings
* A standard classification loss ($\mathcal{L}_{cls}$)

The authors conduct experiments on six datasets spanning finance, criminal justice, healthcare, and image domains. The results show that the method outperforms existing baselines in balancing model utility (accuracy) and various fairness metrics.

**Strengths:**

- The paper tackles a very important and difficult problem: intersectional bias in multi-class scenarios, which has high real-world relevance in high-stakes domains like healthcare (MIMIC).
- As mentioned above, the design of $\mathcal{L}_{intersect}$ and $\mathcal{L}_{FPR}$ are the highlights of this paper, as they are specifically tailored to address this complex problem.
- The ablation study (Table 2) is very effective. It clearly demonstrates that removing any of the MI, Fairness, or FPR components leads to a significant drop in fairness metrics. This proves that each part of the framework is necessary for its final performance.

**Weaknesses:**

- The paper's title and core premise are "Diffusion-Driven," but the paper fails to provide sufficient evidence as to *why* a diffusion model is a necessary component of this framework.
    * The authors claim diffusion models are more robust than VAEs or Autoencoders, but this is just an assertion without a direct comparative experiment.
    * The only encoder-related experiment in the paper is a comparison between $h_1$ and $h_2$ (Table 11). This only proves that $h_1$ is a good choice *within* the diffusion model, but it does **not** prove that the diffusion model itself is superior to others

- The paper's claims regarding hyperparameter tuning raise concerns about its practical utility and reproducibility. The authors rely on a single sensitivity analysis (Table 12), which was **conducted only on the COMPAS dataset** (a binary classification task), to support their claim of "notable stability" (Section 4.4.4). This limited analysis is likely insufficient to demonstrate similar stability for the more complex, multi-class datasets (like MIMIC-III/IV) that are also a key part of the paper's evaluation. This is a significant concern, particularly as the authors also state in the Limitations (Section 5) that the method "requires careful hyperparameter tuning" and mention in the Methods (Section 3.5) their reliance on Bayesian optimization libraries like Hyperopt to search for 5 different $\lambda$ weights.

**Questions:**

1.  Can the authors provide a more direct experiment, or a stronger argument, to justify the superiority of the diffusion model as an encoder? For example, what would the results be if the diffusion encoder and its $\mathcal{L}_{diffusion}$ were replaced with a VAE or others? This is essential to justify the "Diff-Fair" name and the high computational cost of the framework.
2.  Can the authors clarify the apparent contradiction between "notable stability" and "requires careful tuning"? How difficult is tuning in practice (e.g., what is the overhead of the Hyperopt search)? Furthermore, could the authors provide a similar sensitivity analysis for a multi-class dataset like MIMIC-III or MIMIC-IV to validate the model's robustness in more complex scenarios?

---

### Official Review · Reviewer_edwB · 2025-11-01

**Soundness:** 2
**Presentation:** 3
**Contribution:** 2
**Rating:** 4
**Confidence:** 4

**Summary:**

This paper focuses on fair representation learning which aims to address algorithmic bias through a multi-level fairness framework. The trainable framework. The authors claimed that their approach can simultaneously address multiple fairness dimensions.

**Strengths:**

1. The paper proposes an innovative framework that combines diffusion models with fairness constraints to perform representation learning and simultaneously address intersectional bias.
2. The paper demonstrates strong empirical results across multiple benchmarking datasets spanning multiple domains. Crucially, the method consistently outperforms multiple state-of-the-art baselines across these diverse settings.

**Weaknesses:**

1. The paper's justification for using diffusion models relies on theoretical arguments about posterior collapse and trivial reconstructions, but these claims are not theoretically or empirically validated.
2. The paper lacks thorough investigation of computational scalability, particularly for image-based applications. CelebA requires ~30× increase in training time over Colored-MNIST, despite having merely ~3× more samples. This raises concerns about applicability to realistic large-scale visual datasets.
3. The joint optimization of five competing objectives lacks theoretical convergence guarantees. The paper does not establish whether the weighted sum (Eq. 20) converges to a stationary point, or whether different hyperparameter configurations lead to substantially different solutions.
4. The framework requires careful tuning of hyperparameters using Bayesian optimization, with substantially different optimal values across datasets. This limits the method's generalizability and practical applicability.

**Questions:**

1. Can the authors provide ablation studies comparing Diff-Fair with diffusion replaced by VAE or standard autoencoders, to empirically demonstrate that the diffusion architecture is necessary for the fairness improvements observed?
2. Theorem 1 establishes that I(Z;S)=0 implies demographic parity. Can the authors provide visualization of how I(Z;S) changes during training and report its final values across datasets to validate this assumption?

---

### Meta-Review · Area_Chair_VBXJ · 2025-12-19

**Summary:**

After reading the manuscript and reviewers' comment (the authors did not provide their response), I made my recommendation reject. Here are the detailed meta review.

**Research Question**

The authors consider the well-defined fairness learning problem.

**Motivation**

The authors argue that the existing methods struggle to handle intersection bias, i.e., multiple protected attributes.

**Philosophy**

I did not see the philosophy to tackle the above challenges. Instead the authors directly provide their solution.

**Solution**

The authors propose a  framework that addresses intersectional bias through a multi-faceted approach, simultaneously targeting representation-level, group-level, and outcome-level fairness via several complementary mechanisms. However, the rationality of each component is missing.

**Experiments**

The experimental part is not extensive. Several reviewers have concerns on the competitive baselines, ablation study, and the tradeoff between utility and fairness.

**Presentation**

1. Some citation formats are incorrect.

2. A punctuation mark is needed at the end of each equation.

**Summary**

I agree with the reviewers' comments and will not repeat them here. This paper suffers from severe issues, in terms of no philosophy, no rationality of technical part, no strong and valid evaluation.

**Reviewer Concerns:**

Since there is no author response, the original concerns exist.

**Reviewer Scores:**

Since there is no author response, reviewers would not change their scores.

---

### Decision · Program_Chairs · 2026-01-26

Reject